# Mean Value-Amplitude Method for the Determination of Anisotropic Mechanical Properties of Short Fiber Reinforced Thermoplastics

Joachim Hausmann * , Esha , Stefan Schmidt and Janna Krummenacker

Leibniz-Institut für Verbundwerkstoffe (IVW), 67663 Kaiserslautern, Germany; esha@ivw.uni-kl.de (E.); stefan.schmidt@ivw.uni-kl.de (S.S.); janna.krummenacker@ivw.uni-kl.de (J.K.)
* Correspondence: joachim.hausmann@ivw.uni-kl.de

**Abstract:** Short fiber reinforced thermoplastics show distinct anisotropic behaviors due to their microstructure. The mechanical testing of specimens cut from injection molded plates at different angles to the injection molding direction reveals direction-dependent properties. However, these results are an average value for the tested cross section, which in more detail has a core-shell microstructure. When analyzing the stresses and deformation of a structural component, the local anisotropy will be very different compared to these tensile specimens. Therefore, a methodology is needed to transfer the properties obtained by mechanical testing to the local properties of an injection molded component. The core-shell microstructure and tests with different specimen thicknesses enable the determination of microstructure-dependent material properties. This paper presents a method using a mean value representing isotropy and an amplitude applied to the mean value to determine orientation-dependent mechanical properties. The amplitude in turn depends on the degree of anisotropy. The method is applied for extracting the anisotropic Young's modulus of the core and shell layer of short glass fiber reinforced polyamide 46. The results obtained by this method and their reliability are discussed.

**Keywords:** short-fiber composites; injection molding; X-ray computed tomography; mechanical properties

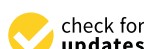



## 1. Introduction

Short fiber reinforced thermoplastics are predestinated for high-volume production, even for parts with complex geometry. Despite short fibers with a length typically between 0.2 and 0.5 mm, a pronounced fiber alignment may result from the injection molding process. The local fiber orientation distribution affects the mechanical as well as the thermomechanical properties throughout the plate [1,2]. To exploit the full capability of the material, the anisotropic behavior must be considered in the design process. A reliable virtual design process becomes increasingly important to save weight and resources. Methods for describing the statistical fiber orientation distribution are already established [3–5]. Besides some possibilities of graphical description, the fiber orientation tensor uses unique numbers to quantify the fiber orientation distribution [4,6,7]. Figure 1 provides some examples, showing the same analyzed specimen. The determination of statistical fiber orientation is possible prior to component design by using suitable process simulation software [6,8–11] or on an existing component by imaging techniques such as X-ray microcomputer tomography (μCT) [2,7]. An advantage of the first method is an affordable prediction, while the second method analyzes the real microstructure and, thus, it is free from deviations by theoretical assumptions. However, analyses of larger volumes by μCT are very costly or even of limited feasibility. After determining the local fiber orientation distribution of a component, it is furthermore necessary to obtain knowledge about the

orientation-dependent material properties. By the µCT image (Figure 1c), the distinct core-shell microstructure becomes visible. The shell layers are created due to the boundary layer formation of material flow near the walls of the tooling, and fibers are aligned preferably in the injection molding direction. The core layer, which is the center part of the material flow between the tooling, is less oriented in comparison to the shell layers with a preferred fiber orientation perpendicular to the molding direction. This becomes visible by the fiber orientation distribution, plotted as a diagram in Figure 1b. The molding of thicker plates leads to thicker core layers, while the thickness of the shell layers is only slightly affected by the plate thickness. Using the eigenvalues of the orientation tensor [4] (Figure 1d), an index of anisotropy (IA) can be determined (Figure 1e). Since the analysis in the following is conducted for an in-plane stress state only, the z component of the orientation tensor is omitted, and the anisotropy is described by the x and y eigenvalues only. It is necessary to consider the component of the examined axis alone, as it describes the statistical amount of fibers aligned in that direction. A value of one for a component in the orientation tensor means that all the fibers are aligned perfectly in that direction, while a value of zero means that all the fibers are aligned perpendicular to that direction. For short fiber reinforcements, the reality is somewhere in between [5].

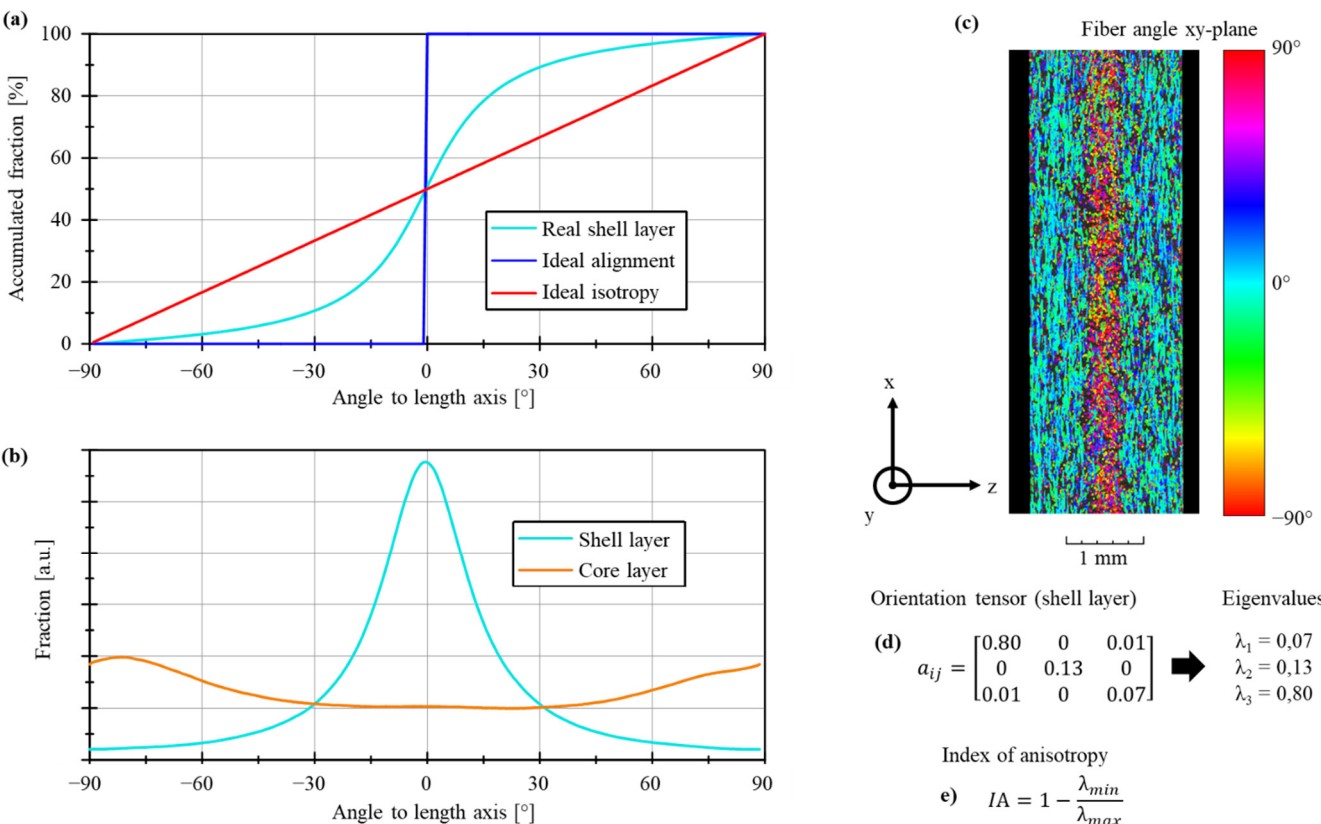

**Figure 1.** Different ways to display the fiber orientation of short fiber reinforced composites: (**a**) accumulated fiber orientation distribution, (**b**) partial fiber orientation distribution, (**c**) colorized microsection by µCT, (**d**) fiber orientation tensor, and (**e**) index of anisotropy determined by the eigenvalues of the orientation tensor [3].

Fu et al. [5] provide analytical equations to estimate mechanical properties by the geometry of the reinforcement and the constituent properties. Promising functions are the Halpin–Tsai equations to calculate the Young's modulus of perfectly aligned short fiber composites in the direction of alignment and perpendicular to that [12,13]. However, very often, the constituent properties and the real microstructure are not exactly known.

The easiest and therefore most common way to determine the mechanical properties of short fiber reinforced thermoplastics experimentally is to mold specimens in a cavity

with the specimen geometry. However, such specimens have a defined microstructure and thus do not allow for the determination of orientation-dependent properties. Therefore, the machining of specimens from injection molded plates in different directions allows for testing in different directions with regard to the molding direction, i.e., the main fiber orientation [2,14]. Nevertheless, properties obtained using machined specimens provide an average value of the material volume within the gauge length, i.e., a pronounced core-shell layer microstructure instead of a distinctive fiber orientation [15,16]. As a consequence, a methodology is required to extract the properties of the material as a function of the anisotropy [17]. Similar approaches are known for endless fiber reinforced composites [18,19]. This allows for a consideration of local material properties in the subsequent design process of a structural component. For this purpose, this paper proposes an efficient method for the determination of the Young's modulus as a function of anisotropy. This can be used to describe local material properties in the process of component design.

## 2. Material and Experiments

Short glass fiber reinforced polyamide 46 (PA46GF15: DSM Stanyl® TW300F3) with a 15% fiber weight fraction was used for specimen preparation. The same material with a 30% fiber weight fraction (PA46GF30: DSM Stanyl® TW300F6) was used for the validation of the results. Plates in the dimension of $80 \times 80$ mm$^2$ were injection molded with two different thicknesses, namely, 2 mm and 3 mm. Tensile specimens according to DIN EN ISO 527-2, Type 1BA were extracted by milling from these plates in such a way that the loading direction of the specimens was parallel (0°) as well as perpendicular (90°) to the molding direction. All the specimens were conditioned to 50% relative humidity prior to testing.

The tensile tests were performed on a universal testing machine (Zwick/Roell, Ulm, Germany) with a displacement of 1 mm/min and a 10 kN load cell. The strain was measured with an extensometer as well as a digital image correlation using the GOM Correlate (GOM GmbH, Braunschweig, Germany). Besides room temperature (RT), tensile tests were conducted at 80 °C within an attached temperature chamber. Details of the materials and specimens, the results of the tensile tests, and their usage for materials modelling were published recently [20].

To analyze the microstructure, cuboids ($7 \times 5 \times$ thickness mm$^3$) were extracted from the center of some specimens. These samples were scanned using an X-ray computer tomograph (nanotom, Phoenix x-ray systems + services GmbH, Wunstorf, Germany) with a resolution of 4 μm. Fiber orientation tensors and the core-shell microstructure of the scanned volumes were determined with the analysis software VG STUDIO MAX (Volume Graphics GmbH, Heidelberg, Germany).

## 3. Resolving Layerwise Properties

This section describes the mathematical procedure to calculate the Young's modulus of the core and the shell layer, respectively. Therefore, an inverse approach is developed using the data from tensile tests and μ-CT. Considering the microstructure of the injection molded plates as a three-layer laminate consisting of a core layer covered by two shell layers, the stiffnesses can be summed up:

$$E = E_{core} \cdot v_{core} + E_{shell} \cdot v_{shell} \tag{1}$$

where $E$ is the measured Young's modulus of the specimen, $E_i$ the local modulus of the layers, and $v_i$ the volume fraction of the layers—here, *core* and *shell*, respectively. Due to the microstructure, the preferred fiber orientation of the shell layers is aligned along the global *x*-axis (molding direction), while the fibers of the core layer are preferably aligned along the global *y*-axis. Taking the notation from the classical laminate theory [21,22], the preferred fiber orientation is the 1-axis in the local coordinate system and the 2-axis perpendicular to the preferred fiber orientation. Taking this into account, Formula (1) can be written as:

$$E_x = E_{2,core} \cdot v_{core} + E_{1,shell} \cdot v_{shell} \tag{2}$$

Figure 2 illustrates the local coordinate systems of the layers and the global coordinate system. The target here is to determine the Young's moduli of the core and shell layer in their local coordinate system in the 1- and 2-directions.

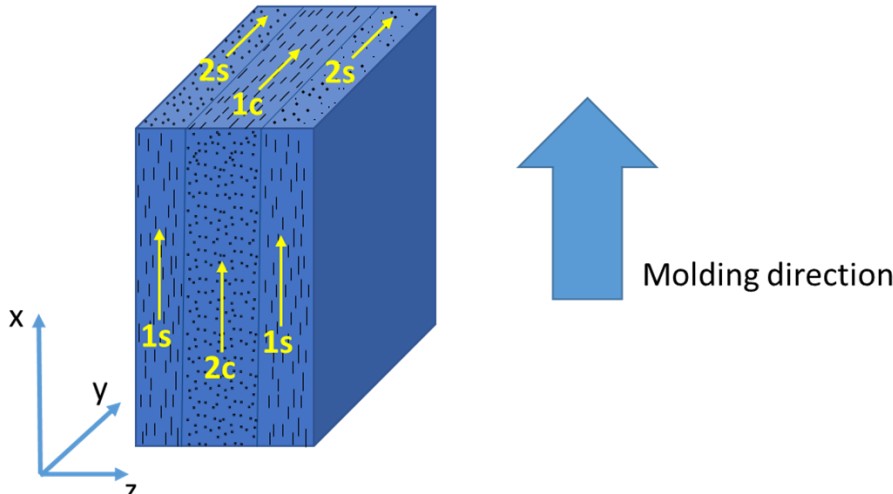

**Figure 2.** Global and local coordinate systems of injection molded plates: x—molding direction, y—transverse to molding direction, z—thickness direction, s—shell layer, c—core layer, 1—mean fiber direction, 2—transverse to mean fiber direction.

To determine the individual Young´s modulus of the core and shell layer, using the results of the tensile tests, some assumptions for simplification need to be made beforehand. Firstly, an isotropic mean value is assumed, which is defined as the average of the global values measured in the 0°- and 90°-directions with specimens cut from plates in different directions. Referring to the global coordinate system, the 0°-direction is parallel to the molding direction and is assumed as the x-axis, while the 90°-direction is the y-axis. Therefore, the isotropic mean value is:

$$E_m = \frac{E_x + E_y}{2} \tag{3}$$

$E$ is the Young´s modulus, and index $m$ denotes the mean value, while $x$ and $y$ are the testing directions in the global coordinate system, as defined before.

Secondly, an amplitude marked by index $a$ applies to the isotropic mean values, which is added for the property in the 1-direction and is subtracted for the property in the 2-direction.

$$E_1 = E_m + E_a \tag{4}$$

$$E_2 = E_m - E_a \tag{5}$$

In analogy to continuous reinforced composites [21,22] the 1-direction is defined as the direction along the preferred orientation of the fibers, and, consequently, the 2-direction is perpendicular in-plane to the 1-direction. Since the degree of anisotropy is different for the core and the shell layer, the amplitude is thus layer-dependent, and Formulas (4) and (5) need to be used for the core and shell layer separately. Furthermore, it is assumed that the core and shell layer are not constraining each other. This means the Poisson ratio is considered to be the same in all directions and layers.

With these assumptions, the layer properties can be summed up with respect to their volume contents $v_{core}$ and $v_{shell}$ to obtain the Young's modulus for the molded plate:

$$E_x = (E_m - E_{a,core}) \cdot v_{core} + (E_m + E_{a,shell}) \cdot v_{shell} \tag{6}$$

with

$$v_{core} = \frac{t_{core}}{t} \tag{7}$$

and

$$v_{shell} = (1 - v_{core}) \tag{8}$$

$t_{core}$ is the thickness of the core layer, while $t$ denotes the total thickness of the specimen.

The two unknowns in Formula (6) are the amplitudes for the core and the shell layer, respectively. Since the volume ratios of the core and shell layer depend on the thickness of the specimens, an equation system can be formulated and resolved to the amplitudes of the core and shell, leading to:

$$\sigma_{a,core} = k_a \left( \sigma_m - \frac{\sigma_{x,t1} \cdot v_{shell,t2} - \sigma_{x,t2} \cdot v_{shell,t1}}{v_{core,t1} \cdot v_{shell,t2} - v_{core,t2} \cdot v_{shell,t1}} \right) \tag{9}$$

and

$$\sigma_{a,shell} = \frac{\sigma_{x,t2} - v_{core,t2} (\sigma_m - \sigma_{a,core})}{v_{shell,t2}} - \sigma_m \tag{10}$$

Here, index $t1$ denotes the data of the thinner plates, while index $t2$ denotes the data of the thicker plates. $k_a$ is a correction factor for the amplitude, which is determined as 0.5 to avoid unrealistically high amplitudes. As input for Equations (9) and (10), the volume of the core or shell layer needs to be measured from microscopic cross sections or μCT analysis. Mechanical tests in the x- and y-directions are necessary to provide the mean value $E_m$ and the values in the x-direction of both thicknesses: $E_{x,t1}$ and $E_{x,t2}$. Thus, specimens from the plates with two different thicknesses tested in two different directions are necessary to provide the input data for the equations.

## 4. Results

### 4.1. μCT-Analysis

μCT-scans were used to distinguish the core and shell layers and to determine the fiber orientation tensors of both. Exemplary cross sections of the μCT scans and the anisotropy values are provided by Figure 3. The eigenvalues $\lambda_x$ and $\lambda_y$ are the averages from two specimens. The values given of the shell layers are the average of the left and right shell layers. It can be seen that the fiber orientation distribution is similar for both thicknesses, while it is more anisotropic for the skin layer than it is for the core layer. This results in a more pronounced anisotropy (full thickness) for the thinner specimen. Both fiber fractions show a similar behavior. The thickness of the shell layers is 0.81 mm for the 2 mm specimens and 0.92 mm for the 3 mm specimens. This leads to volume fractions of 0.19 and 0.388 for the core layer of the 2 mm and 3 mm plates, respectively, for both fiber fractions, despite the fact that the transition between the core and shell layers is more pronounced for the higher fiber fraction.

### 4.2. Tensile Tests

The exemplary stress-strain curves obtained by the tensile tests are displayed in Figure 4. As expected, higher temperatures yield lower stresses and stiffnesses along with higher fracture strains. Since the volume fraction of the more anisotropic shell layers is higher for the 2 mm thick specimens, the anisotropic behavior is more pronounced compared to that of the 3 mm specimens. This results in higher values for the stiffness and strength of the 0°-specimens and lower values for the 90°-specimens with 2 mm thickness compared to those for the respective 3 mm thick specimens. This behavior meets the expectations. The Young´s modulus was determined as the secant modulus in the strain range of 0.05 to 0.25% according to DIN EN ISO 527-1. The Young´s moduli of all the specimens are summarized in Table 1 along with their standard deviation.

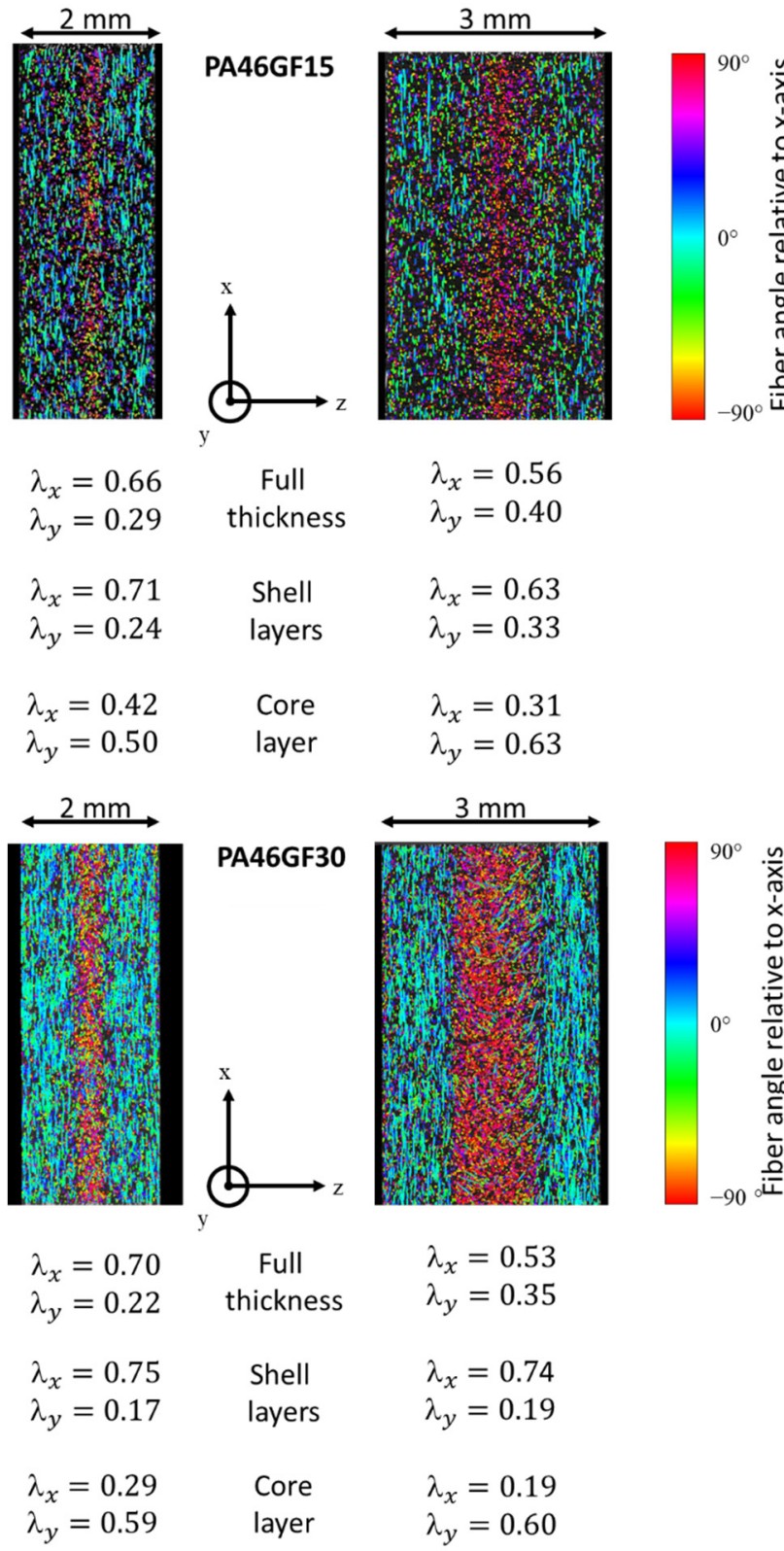

**Figure 3.** μCT-scans of 2 mm specimens (**left**) and 3 mm specimens (**right**) with a 15% fiber fraction (**top**) and 30% fiber fraction (**bottom**) and determined eigenvalues of the fiber orientation tensors.

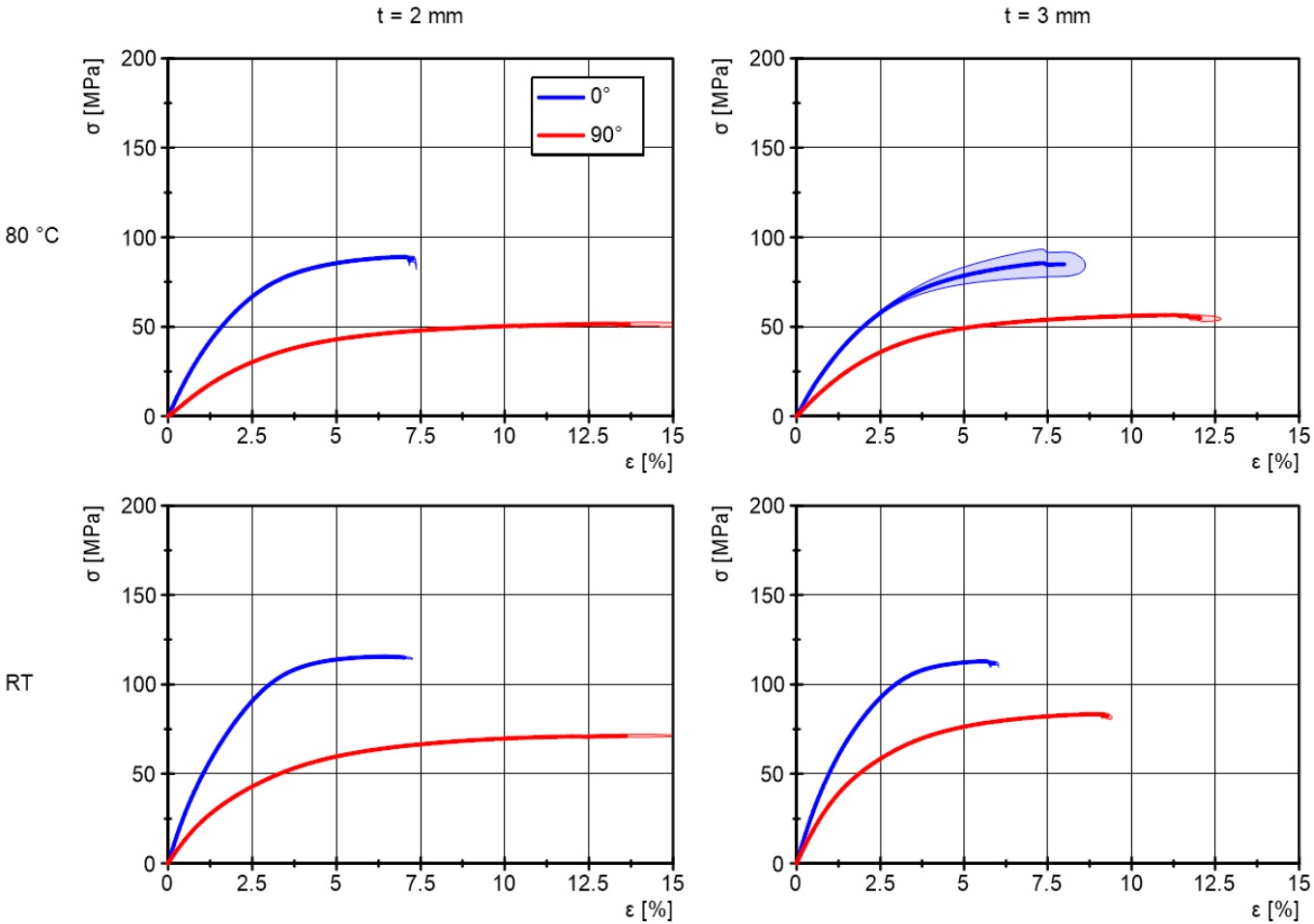

**Figure 4.** Stress-strain curves for the 30% fiber weight content of 2 mm thick specimens (**left**) and 3 mm thick specimens (**right**) at room temperature (**bottom**) and 80 °C (**top**). Curves display the average of five specimens, and the envelope marks the standard deviation.

**Table 1.** Engineering values (exemplary for the 15% fiber weight content) from the tensile tests; average of five specimens tested for each parameter set with the standard deviation.

| Temperature | Orientation | Thickness [mm] | Young's Modulus [MPa] | Tensile Strength [MPa] | Strain at Break [%] |
|---|---|---|---|---|---|
| RT | 0° | 2 | 3249 ± 36 | 87.4 ± 1.1 | 14.34 ± 1.42 |
| | | 3 | 2992 ± 67 | 83.3 ± 0.6 | 15.16 ± 0.64 |
| | 90° | 2 | 1988 ± 79 | 69.4 ± 0.4 | 21.09 ± 2.0 |
| | | 3 | 2296 ± 77 | 74.0 ± 0.7 | 17.88 ± 1.49 |
| 80 °C | 0° | 2 | 2627 ± 238 | 72.5 ± 1.2 | 13.69 ± 1.19 |
| | | 3 | 2092 ± 137 | 69.9 ± 7.7 | 14.06 ± 0.56 |
| | 90° | 2 | 1292 ± 69 | 49.8 ± 1.0 | 22.86 ± 2.11 |
| | | 3 | 1356 ± 55 | 53.1 ± 0.2 | 20.39 ± 0.97 |

*4.3. Analytical Results*

With the volume fractions of the core and shell layer determined by μCT and the Young´s modulus obtained from the tensile tests, the mean value and amplitudes of the core and shell layer can be derived by Formulas (3), (9) and (10). For the mean value, the average of the moduli of both thicknesses and both directions were taken. With these, the properties in the material coordinate system (1–2-system) are obtained by Formulas (2) and (3). A summary of the results is given in Table 2.

**Table 2.** Analytical results of the anisotropic modulus in the MPa of the core and shell layers in the material coordinate system.

| Fiber Content | 15% | | 30% | |
|---|---|---|---|---|
| Temperature | RT | 80 °C | RT | 80 °C |
| $E_m$ | 2631 | 1842 | 4662 | 2775 |
| $E_{a,core}$ | 212 | 691 | 1051 | 1064 |
| $E_{a,shell}$ | 727 | 852 | 3026 | 1723 |
| $E_{1,core}$ | 2843 | 2533 | 5712 | 3838 |
| $E_{1,shell}$ | 3358 | 2694 | 7688 | 4498 |
| $E_{2,core}$ | 2420 | 1151 | 3611 | 1711 |
| $E_{2,shell}$ | 1905 | 990 | 1636 | 1052 |

## 5. Discussion

The tendency of the layerwise-determined modulus is plausible in general. The shell layers show higher anisotropy than the core layers, correlating well with the differences in the degree of anisotropy due to the microstructure. Furthermore, a higher anisotropy is obtained for higher testing temperatures. This observation is also plausible since the matrix is much more temperature sensitive compared to the fibers and, thus, the anisotropy of the composite will increase with increasing temperature.

It is assumed by Fu et al. [5] that there is a linear relationship between the Young's modulus and the eigenvalue of the fiber orientation. Therefore, Figure 5 correlates the modulus of the tensile test on the layered specimens as well as the results of the layerwise extraction by the mean value-amplitude method to the individual eigenvalues. In general, a linear relationship is visible. However, single points in the diagrams deviate a lot from the linear approximation. Especially, when test results with inconsistent relations to each other are used as the input for the layerwise extraction, a big deviation between the experimental and analytical curve is obtained. This can be seen by the curves for room temperature and 30% fiber content (green curves at the bottom of Figure 5). Here, the experimental modulus in the x-direction of the 2 mm specimens is lower than that of the 3 mm specimens; both are within the range of one standard deviation. Since this behavior is physically not evident, a high deviation of the experimental and analytical results is obtained. The reasons for the failure of the experimental results at room temperature may be that the testing temperature is very close to the glass transition temperature of PA46 in the conditioned state, and small temperature deviations have a big impact on the mechanical response. Since the linear behaviors of the experimental and layerwise extraction are very close to each other, it may be most reliable to use the experimental and extracted values for a common approximation.

Another considerable parameter is the initial definition of the core and shell layers. In Figure 6, the orientation tensor components xx and yy are plotted along the specimen's thickness. It is evident that not a sudden frontier but rather a transition zone exists between the core and shell layers. The exact selection of this frontier can cause non-negligible shifts in the volume fractions of the layers. Additionally, especially when the core layer is very thin, the orientation tensor is prone to the influence of the transition zone depending on the selected frontier. Since the analytical results are very sensitive to the volume fraction of the core layer, this effect might be significant. Especially for the specimens with a lower fiber content, the transition is hard to distinguish. The adjustment of the correction factor $k_a$ in Formula (9) may be used to compensate such deviations.

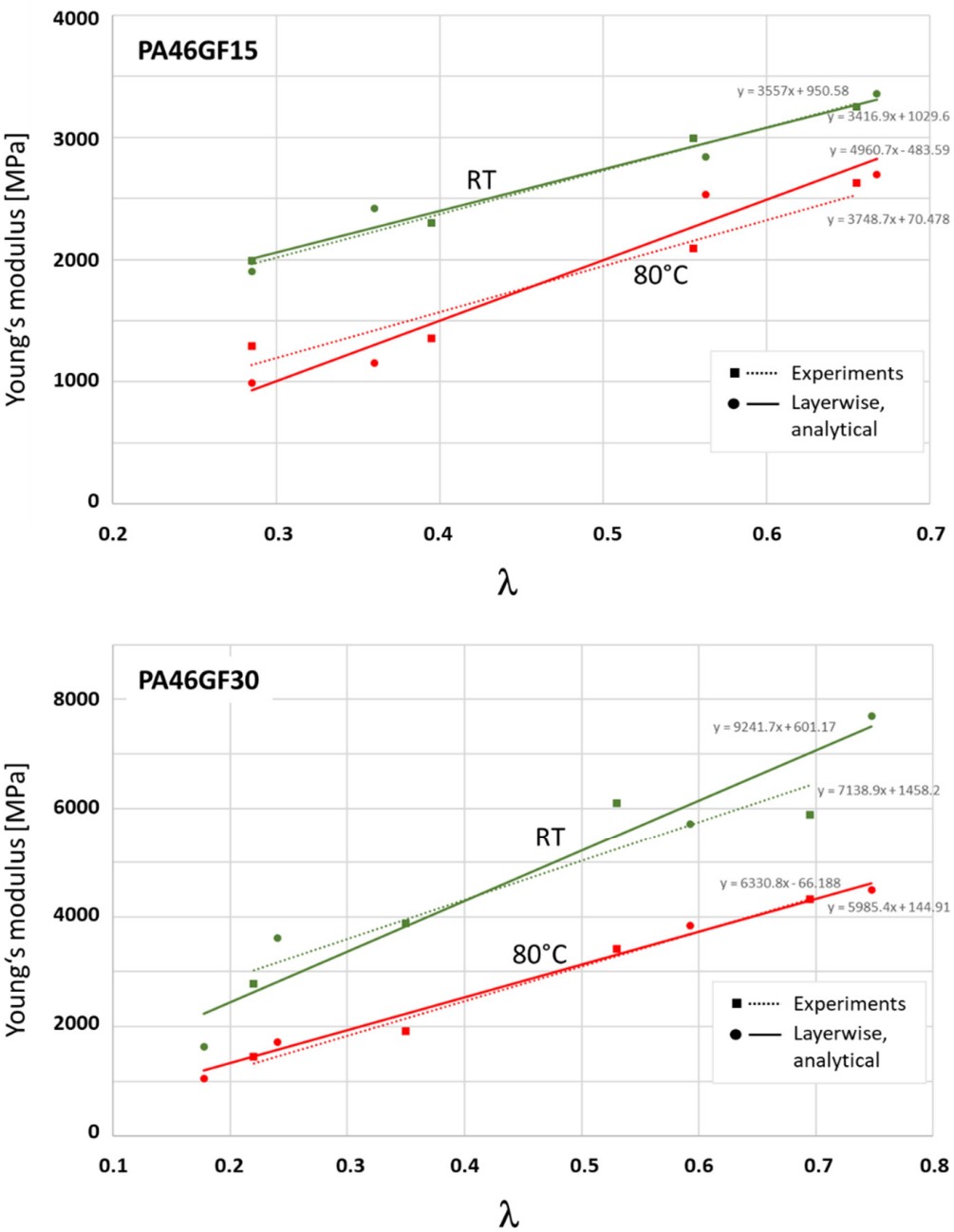

**Figure 5.** Experimentally determined Young's modulus of the specimens and the layerwise analytically resolved Young's modulus versus the eigenvalue λ of the fiber orientation tensor for a 15% fiber weight content (**top**) and 30% fiber weight content (**bottom**). Each modulus is related to the individual eigenvalue, which is valid for the same direction. Equations for linear approximation are given in the diagrams.

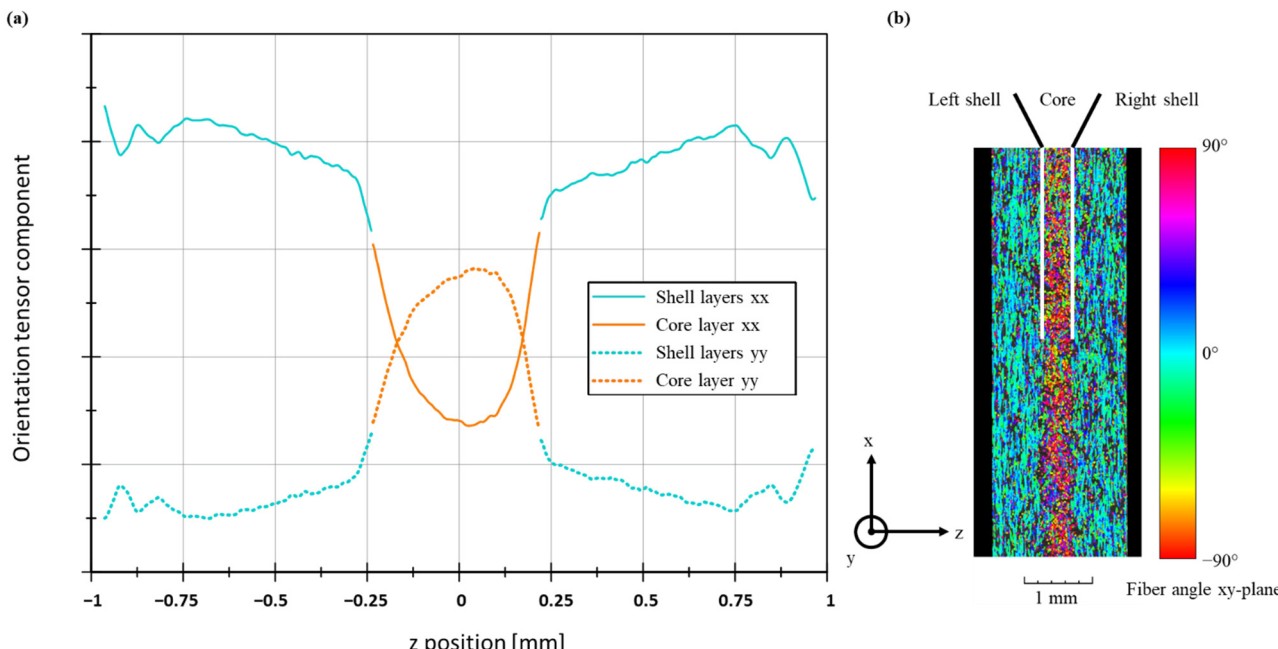

**Figure 6.** (**a**) Orientation tensor components xx and yy along the specimen's thickness and (**b**) µCT cross section of the specimen, both with indicated defined frontiers between the core and shell layers.

## 6. Conclusions

For the reliable stress and deformation analysis of short fiber reinforced thermoplastic components, knowledge about the local anisotropic material properties is necessary. Since material testing on the coupon level provides averaged values of the tested volume with three distinct layers, an extraction method is required to resolve the fiber orientation-dependent properties. Therefore, a mean value amplitude method is proposed to determine the Young's modulus as a function of the eigenvalues of the fiber orientation tensor. A combination of experimentally obtained and analytically extracted moduli may lead to the most reliable approximation.

The proposed method may be useful for the strength behavior, too. However, the determination of layerwise strength values by that method is limited by the fact that the overall strength of the specimen is limited by the failure of the first layer, since it cannot be expected that all the layers of the tested volume fail at the same strain. Therefore, the extracted strength values are not as reliable as the modulus values are. Nevertheless, the first attempts in that direction are promising.

In summary, the proposed method to extract layerwise properties using tensile test data from specimens tested in two different directions and with two different thicknesses reveals a relationship between anisotropy and mechanical properties. This relationship can be used for the detailed finite element analysis of injection molded components with consideration of the local material properties. More validation is necessary to establish the method in the design process.

**Author Contributions:** Conceptualization, J.H.; methodology, J.H.; investigation, E., S.S. and J.K.; data curation, S.S.; writing—original draft preparation, J.H.; writing—review and editing, E., S.S. and J.K.; visualization, J.H. and S.S.; supervision, J.H.; project administration, J.H.; funding acquisition, J.H. and J.K. All authors have read and agreed to the published version of the manuscript.

**Funding:** This research received no external funding.

**Data Availability Statement:** The experimental database used as the input data for the mean value-amplitude method is the same as that used for the development of the multi-parameter model described in [20].

**Acknowledgments:** The authors would like to acknowledge Hermann Giertzsch and Werner Gölzer from Leibniz-Institut für Verbundwerkstoffe for their excellent support by conducting μCT, tensile tests, and specimen preparation.

**Conflicts of Interest:** The authors declare no conflict of interest.

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
