# Peer review of "Mean Value-Amplitude Method for the Determination of Anisotropic Mechanical Properties of Short Fiber Reinforced Thermoplastics"

_jcs, doi:10.3390/jcs6060179_

Round 1

Reviewer 1 Report

The article covers a very interesting topic.

However, the article needs to be supplemented and corrected.

Literature review should be improved, more references related to the topic should be completed.

The article should be corrected in accordance with the guidelines (for example, replace commas with dots in the data in the figures 5, 6; spaces and other symbols).

Figure 6, should be marked with figures a, b, c, d (and so is figure 3 a, b).

There is no exact description of the properties tested (theoretical considerations).

The article may be published after completion and correction.

Author Response

Dear Reviewer,

Thank you very much for your valuable comments which helped to improve the paper. Please find below our response.

Literature review should be improved, more references related to the topic should be completed. => Several additional references were added.

The article should be corrected in accordance with the guidelines (for example, replace commas with dots in the data in the figures 5, 6; spaces and other symbols). => Fig. 5 and 6 were corrected, symbols were checked for consistency

Figure 6, should be marked with figures a, b, c, d (and so is figure 3 a, b). => done

There is no exact description of the properties tested (theoretical considerations). => I do not understand what you mean here?

Reviewer 2 Report

The authors use a rule of mixtures to model the contribution of the layer and the core regions of a mold injected specimen to its Young’s modulus. The approach is elegant and has appealing for the researchers.

I agree with the authors on the fact that there is a transition are from the core to the shell region. Here the formulae can include a fuzzy factor to add possible deviations to the values provided by the formula.

My congratulations to the authors for their work.

Author Response

Dear Reviewer,

Thank you very much for your valuable comments which helped to improve the paper. Please find below our response.

I agree with the authors on the fact that there is a transition are from the core to the shell region. Here the formulae can include a fuzzy factor to add possible deviations to the values provided by the formula. => on page 9 the sentence "Adjustment of the correction factor ka in formula 9 may be used to compensate such deviations" has been added.

My congratulations to the authors for their work => Thank you very much!

Reviewer 3 Report

It would be interesting to compare with a greater number of references  

Author Response

Dear Reviewer,

Thank you very much for your comment. We added several references in the introduction.